# Tomentosin Displays Anti-Carcinogenic Effect in Human Osteosarcoma MG-63 Cells via the Induction of Intracellular Reactive Oxygen Species

**DOI:** 10.3390/ijms20061508

**Published:** 2019-03-26

**Authors:** Chang Min Lee, Jongsung Lee, Myeong Jin Nam, Youn Soo Choi, See-Hyoung Park

**Affiliations:** 1Department of Biological Science, Gachon University, Seongnam 13120, Korea; yycc456@naver.com; 2Department of Integrative Biotechnology, Sungkyunkwan University, Suwon 16419, Korea; bioneer@skku.edu; 3Department of Biomedical Sciences, Seoul National University Graduate School, Department of Medicine, College of Medicine, Seoul National University, Seoul 03080, Korea; 4Department of Bio and Chemical Engineering, Hongik University, Sejong 30016, Korea

**Keywords:** osteosarcoma, tomentosin, reactive oxygen species (ROS)

## Abstract

Tomentosin is a natural sesquiterpene lactone extracted from various plants and is widely used as a medicine because it exhibits essential therapeutic properties. In this study, we investigated the anti-carcinogenic effects of tomentosin in human osteosarcoma MG-63 cells by performing cell migration/viability/proliferation, apoptosis, and reactive oxygen species (ROS) analysis assays. MG-63 cells were treated with various doses of tomentosin. After treatment with tomentosin, MG-63 cells were analyzed using the MTT assay, colony formation assay, cell counting assay, wound healing assay, Boyden chamber assay, zymography assay, cell cycle analysis, FITC Annexin V apoptosis assay, terminal deoxynucleotidyl transferase dUTP nick end labeling assay, western blot analysis, and ROS detection analysis. Our results indicated that tomentosin decreased cell viability and migration ability in MG-63 cells. Moreover, tomentosin induced apoptosis, cell cycle arrest, DNA damage, and ROS production in MG-63 cells. Furthermore, tomentosin-induced intracellular ROS decreased cell viability and induced apoptosis, cell cycle arrest, and DNA damage in MG-63 cells. Taken together, our results suggested that tomentosin exerted anti-carcinogenic effects in MG-63 cells by induction of intracellular ROS.

## 1. Introduction

Osteosarcoma, also known as bone tumor, generally affects children and adolescents [1]. According to recent research, every year, approximately 1150 new cases of high-grade osteosarcoma are detected in the European Union [2]. Moreover, according to the latest cancer statistics, osteosarcoma was ranked as having the lowest five-year relative survival rate among the most common childhood and adolescent cancers in the United Sates [3]. Since modern therapeutic methods have developed rapidly, the outcome for patients with osteosarcoma has improved [4]. Typical treatments for osteosarcoma include surgery and intensive chemotherapy [5]. However, with respect to recurrent and metastatic osteosarcoma, previous treatments, including surgery and chemotherapy exhibit limited therapeutic effect [6]. Therefore, it is necessary to develop new therapeutic strategies to treat osteosarcoma. 

In modern medicine, natural compounds have been a crucial source of many drugs. Tomentosin is a sesquiterpene lactone isolated from *Inula viscosa* (L.) Aiton., which is widely used as a medicine because it exhibits essential therapeutic properties, such as anti-inflammatory [7], anti-bacterial, and anti-cancer activities [8,9]. Moreover, recent research has shown that tomentosin has an anti-proliferative effect on human cancer cell lines in vitro. Tomentosin also induces apoptosis via telomere shortening in human cervical cancer cells [9]. 

Tomentosin has been shown to have an anti-carcinogenic effect in human melanoma cells, despite the fact that malignant melanoma is an aggressive tumor resisting frequent chemotherapy [10]. However, the anti-cancer effects of tomentosin in various cancer cell lines have barely been investigated, especially in osteosarcoma. Therefore, the present study investigated the anti-carcinogenic effects of tomentosin in human osteosarcoma MG-63 cells.

Reactive oxygen species (ROS) are generated during the process of mitochondrial oxidative metabolism as well as in response to cellular stress [11]. It has been reported that ROS function as crucial chemical messengers and play an important role in cell growth and proliferation [12]. Generally, the anti-carcinogenic characteristic of phytochemicals is believed to be associated with their ability to suppress intracellular ROS [13]. However, the pro-oxidant activity of phytochemicals, rather than their anti-oxidant activity in cancer cells, has been reported to be a crucial mechanism for mediating their anti-carcinogenic activities [14]. Celastrol has been shown to induce G2/M phase cell cycle arrest, apoptosis, and autophagy through the ROS/Jun N-terminal kinase (JNK) signaling pathway in human osteosarcoma cells [15]. Moreover, phenyl arsine oxide was shown to induce apoptosis in human hepatocellular carcinoma HepG2 cells via ROS-dependent signaling pathways [16]. 

The aim of our study was to evaluate the anti-carcinogenic effects of tomentosin in human osteosarcoma MG-63 cells. We investigated the mechanisms of tomentosin-induced cell death in MG-63 cells.

## 2. Results

### 2.1. Tomentosin Inhibited Proliferation and Induced G2/M Cell Cycle Arrest in MG-63 Cells

MG-63 cells were treated with different concentrations of tomentosin (0, 10, 20, and 40 μM) dissolved in dimethyl sulfoxide (DMSO) (final concentration of 0.1%) for 24 and 48 h. The structure of tomentosin is shown in Figure 1a. We observed detectable morphological changes after treatment of MG-63 cells with tomentosin (Figure 1b). After treatment with tomentosin, the MTT assay was performed. As shown in Figure 1c, the viability of MG-63 cells was decreased after tomentosin treatment in a dose- and time-dependent manner. The IC_50_ (concentration that inhibits 50% of cell survival) value of tomentosin in MG-63 cells was approximately 40 µM after 24 h of treatment. In addition, cell counting assay results showed that the number of cells was significantly decreased after treatment with 20 and 40 µM of tomentosin for 24 and 48 h (Figure 1d). Similarly, clonogenic survival of MG-63 cells was markedly decreased when treated with 10 µM of tomentosin compared to control group (Figure 1e). We counted the number of colonies and the data were analyzed statistically (Figure 1f). Taken together, our results indicated that tomentosin inhibited both the proliferation and clonogenic survival of MG-63 cells. To evaluate the effects of tomentosin on the cell cycle, a cell cycle assay was performed. MG-63 cells were treated with different concentrations of tomentosin (0, 10, 20, and 40 μM) for 48 h and analyzed using flow cytometry. Cell cycle analysis results showed a dose-dependent effect of tomentosin on the cell cycle in MG-63 cells (Figure 1g). After 48 h of treatment with 40 µM tomentosin, the percentage of cells in the G2/M population increased from 25.24 to 49.53%. The bar graph shows a significant increase in the proportion of cells in the G2/M phase as compared to that in the control group (Figure 1h). Our results demonstrated that tomentosin exerted anti-proliferative effects through cell cycle arrest in the G2/M phase. 

### 2.2. Tomentosin Inhibited Migration and Invasion of MG-63 Cells

To determine the effects of tomentosin on cell migration, a wound healing assay was performed in MG-63 cells. Migration of tomentosin-treated MG-63 cells was significantly decreased in a dose- and time-dependent manner (Figure 2a). The results showed that the wound area was healed perfectly after 36 h in the control group. Statistical analysis showed that the empty area was significantly increased in tomentosin-treated MG-63 cells compared to that in the control group (Figure 2b). Furthermore, we performed Boyden chamber assays to evaluate the invasive ability of MG-63 cells after tomentosin treatment (Figure 2c). We found that the number of invading cells was significantly decreased after treatment with tomentosin (Figure 2d). We also performed zymography analysis to evaluate the activity of matrix metalloproteinase (MMP)-2 in tomentosin-treated MG-63 cells (Figure 2e). The enzymatic activity of MMP-2 in MG-63 cells was significantly inhibited after treatment with tomentosin (Figure 2f). Taken together, our results indicated that tomentosin reduced the migration and invasion ability of MG-63 cells.

### 2.3. Tomentosin Induced Apoptosis in MG-63 Cells

To investigate tomentosin-induced apoptosis in MG-63 cells, we next performed an annexin V/propidium iodide (PI) double-staining assay. We stained MG-63 cells with annexin V and PI dye after treatment with different concentrations of tomentosin (0, 10, 20, and 40 μM) for 24 and 48 h. After treatment, the cells were analyzed using flow cytometry. Our results indicated that the levels of annexin V^+^/PI^–^ cells and annexin V^+^/PI^+^ cells were increased in tomentosin-treated MG-63 cells and the bar graphs showed that the percentage of total apoptotic cells (early and late apoptosis) was increased in MG-63 cells treated with tomentosin in a dose- and time-dependent manner (Figure 3a–d). Furthermore, to observe tomentosin-induced DNA fragmentation in the nuclei of MG-63 cells, the terminal deoxynucleotidyl transferase (TdT) dUTP nick-end labeling (TUNEL) assay was performed after treatment of MG-63 cells with different concentrations of tomentosin (0, 10, 20, and 40 μM) for 24 and 48 h. We observed that green fluorescence (a sign of DNA fragmentation) was increased in tomentosin-treated MG-63 cells in a dose- and time-dependent manner (Figure 3e–h). Merged images showed the DNA-fragmented nuclei of MG-63 cells treated with tomentosin. Thus, our results indicated that tomentosin induced apoptosis and DNA fragmentation in the nuclei of MG-63 cells. To investigate the tomentosin-induced apoptosis signaling pathways in MG-63 cells, we performed western blot analysis. We analyzed the protein levels of caspase-3, caspase-7, caspase-8, caspase-9, activated caspases, poly (ADP-ribose) polymerase (PARP), cleaved PARP, H2AX, γH2AX, Bcl-2, Bcl-xl, and FOXO3 after treatment of MG-63 cells with tomentosin (0, 10, 20, and 40 μM) for 48 h (Figure 3i). We found that the tomentosin treatment decreased the levels of caspase-3, caspase-7, caspase-8, and caspase-9, and also induced PARP cleavage. Furthermore, we analyzed the protein levels of Akt, phospho (p)Akt, JNK, pJNK, p38, pp38, extracellular signal-regulated kinase (ERK), pERK, peroxiredoxin-1, and Bax after treatment of MG-63 cells with tomentosin. There was no significant change in the levels of JNK, pJNK, p38, and pp38. However, we found that tomentosin treatment decreased peroxiredoxin-1 and pERK levels and increased γH2AX levels in MG-63 cells. We also found that the Bax and p27 expression levels were significantly increased after treatment with tomentosin. Taken together, our results showed that tomentosin treatment induced apoptosis in MG-63 cells via activation of caspases and mitochondrial pro-apoptotic proteins. To verify the role of FOXO3 for G2/M cell cycle arrest in MG-63 cells treated with tomentosin, we performed Western blotting analysis of FOXO3 and p27 expression level in MG-63 cells transfected with control siRNA or FOXO3 siRNA followed by tomentosin treatment. As shown in Figure 3j, after tomentosin treatment for 24 h, compared to the DMSO control, both of the FOXO3 and p27 expression levels were increased in MG-63 cells transfected with control siRNA but neither the FOXO3 nor p27 expression levels were changed in MG-63 cells transfected with siRNA against FOXO3. This result suggests that tomentosin may induce p27-mediated G2/M cell cycle arrest in a FOXO3-dependent manner.

### 2.4. Tomentosin Increased Intracellular ROS Level in MG-63 Cells

Since the expression level of peroxiredoxin-1 was downregulated in MG-63 cells after the treatment with tomentosin, we measured the ROS level in MG-63 cells treated with tomentosin using a 2’,7’-dichlorofluorescin diacetate (DCF-DA) assay. The cells were treated with different concentrations of tomentosin for 48 h. Flow cytometry analysis showed that mean fluorescence intensity (MFI) values of the cells treated with 0, 10, 20, and 40 μM of tomentosin were 125, 191, 221, and 449, respectively (Figure 4a,b). We also used N-acetyl cysteine (NAC) to investigate the effect of intracellular ROS. The cells were treated with 40 μM of tomentosin or/and 10 mM of NAC for 48 h. Flow cytometry results also showed that the MFI values of control, NAC-treated cells, cells treated with tomentosin, and cells treated with both were 151, 126, 716, and 149, respectively (Figure 4c,d). 

### 2.5. Tomentosin-Induced Intracellular ROS Inhibited Proliferation and Induced G2/M Cell Cycle Arrest in MG-63 Cells

To investigate the effects of tomentosin-induced intracellular ROS in MG-63 cells, the cells were treated with 10 mM of NAC and 40 μM of tomentosin for 48 h. Compared with the tomentosin-treated group, there were a greater number of viable cells in the NAC + tomentosin-treated group (Figure 5a). We measured the viability of control cells, NAC-treated cells, cells treated with 40 μM of tomentosin, and cells treated with both using the MTT assay. The results showed that the viability of tomentosin-treated cells was lower than that of cells treated with NAC + tomentosin (Figure 5b). These results suggested that the tomentosin-induced increase in intracellular ROS decreased the viability of MG-63 cells. The cell counting assay also revealed that tomentosin induced ROS-mediated inhibition of proliferation of MG-63 cells (Figure 5c). To evaluate the effects of tomentosin-induced ROS on cell cycle, a cell cycle assay was performed (Figure 5d,e). The proportion of cells in the G2/M phase was significantly higher when treated with tomentosin alone as compared to when cells were treated with NAC + tomentosin. These results indicated that tomentosin-induced intracellular ROS mediated cell cycle arrest in MG-63 cells at the G2/M phase.

### 2.6. Tomentosin-Induced Intracellular ROS Inhibited Migration and Invasion of MG-63 Cells

To determine the effects of tomentosin-induced intracellular ROS on cell migration, Boyden chamber and zymography assays were performed on MG-63 cells. The cells were treated with 10 mM of NAC and/or 40 μM of tomentosin for 48 h. We found that the number of invading cells was significantly lower in the tomentosin-treated group compared with NAC + tomentosin-treated group (Figure 6a,b). These results suggested that tomentosin-induced intracellular ROS decreased the invasion ability of MG-63 cells. We also performed zymography analysis to evaluate the activity of MMP-2 in NAC- and/or tomentosin-treated MG-63 cells (Figure 6c,d). The enzymatic activity of MMP-2 in tomentosin-treated cells was lower compared with that in NAC + tomentosin-treated cells. 

### 2.7. Tomentosin-Induced Intracellular ROS Promoted Apoptosis in MG-63 Cells

To investigate apoptosis caused by tomentosin-induced intracellular ROS in MG-63 cells, we performed an annexin V/PI double-staining assay, TUNEL assay, and Western blotting. The cells were treated with 10 mM NAC and/or 40 μM tomentosin for 48 h. FACS analysis showed that the total apoptotic cell rates of tomentosin-treated cells were higher compared with those of NAC + tomentosin-treated cells (Figure 7a,b). These results suggested that tomentosin-induced intracellular ROS caused apoptosis of MG-63 cells. Moreover, a TUNEL assay was performed to study the effects of tomentosin-induced intracellular ROS in MG-63 cells. We observed that green fluorescence (indicative of apoptosis) was higher in tomentosin-treated cells compared with that in NAC + tomentosin-treated cells (Figure 7c,d). These results showed that tomentosin-induced intracellular ROS promoted DNA damage in MG-63 cells. To investigate the apoptotic signaling pathways activated by tomentosin-induced intracellular ROS in MG-63 cells, we performed Western blot analysis. We analyzed the protein levels of caspase-3, caspase-7, caspase-8, caspase-9, PARP, cleaved PARP, H2AX, γH2AX, Bcl-xl, FOXO3, AKT, pAKT, JNK, pJNK, p38, pp38, ERK, pERK, Bcl-2, and Bax (Figure 7e). We observed that expression levels of caspase-3, caspase-7, caspase-8, and caspase-9 were lower in tomentosin-treated cells compared with those in NAC + tomentosin-treated cells. We also found that expression levels of cleaved PARP, γH2AX, and Bax were higher in tomentosin-treated cells compared with those in NAC + tomentosin-treated cells. These results suggested that tomentosin-induced intracellular ROS activated the apoptotic signaling pathways in MG-63 cells.

## 3. Discussion

Osteosarcoma is a highly prevalent cancer worldwide, commonly occurring in children and adolescents [17]. Although the prognosis of osteosarcoma has improved because of the development of new therapeutic methods, the long-term survival rate for osteosarcoma has stagnated [15]. Therefore, it is necessary to develop novel innovative drugs to improve long-term outcome in patients with osteosarcoma. The objective of the present study was to identify effective anti-carcinogenic agents for human osteosarcoma.

Tomentosin is a natural compound extracted from *I. viscosa* L. and it is recognized as an effective therapeutic agent against various cancers [9]. However, the anti-carcinogenic effects of tomentosin on many kinds of cancer have not been studied so far. In this study, we examined the anti-carcinogenic effects of tomentosin on human osteosarcoma MG-63 cells. We selected the MG-63 osteosarcoma cell line for this specific study because we wanted to investigate the possibility of tomentosin as a novel therapeutic option for p53 null osteosarcoma patients. MG-63 cells are known to have no functional p53 (p53 null) and thus serve as a good model cell line for the development of novel therapeutic treatments for osteosarcoma patients with p53 null status. We found that tomentosin inhibited the proliferation and migration of MG-63 cells. Moreover, our results also showed that tomentosin induced apoptosis, DNA fragmentation, cell cycle arrest, and activation of apoptotic signaling pathways in MG-63 cells. We also observed that tomentosin induced intracellular ROS in MG-63 cells. 

In this study, we investigated the anti-carcinogenic effect of tomentosin in human osteosarcoma MG-63 cells. When cells were treated with a tomentosin dose higher than 40 µM, the cytotoxic effect was too strong and the cells almost died. Therefore, 40 µM was the maximum concentration of tomentosin used in this study.

When cells were treated with relatively low concentrations of tomentosin (0, 10, and 20 µM), the apoptotic rates were low. However, a low dose of tomentosin significantly decreased cell migration (migration assay) and invasion (invasion assay) abilities. Moreover, gelatinase activity was significantly decreased in the 20 µM tomentosin-treated group. These results suggested that tomentosin suppressed migration and invasion abilities under conditions of low concentration of tomentosin. When the cells were treated with 40 µM of tomentosin, however, the proportion of TUNEL-positive cells was much higher than that in other groups. These results suggested that tomentosin induced apoptosis when the cells were treated with high concentrations of tomentosin. Taken together, our results suggested that tomentosin suppressed migration/invasion ability and subsequently induced apoptosis in MG-63 cells depending on the treatment dose.

The PI3K/Akt signaling pathway has been studied widely for its roles in the cell survival process [18]. It was previously reported that tamoxifen induced apoptosis, and the pAkt expression level was significantly decreased after tamoxifen treatment in various kinds of cells, such as MDA-MB-231, MDA-MB-468, MDA-MB-453, and SK-BR-3 cells [19]. However, our results indicated that the pAkt expression level was not changed significantly in MG-63 cells after treatment with tomentosin. JNKs play a crucial role in apoptotic signaling pathways [20]. A recent study showed that hesperitin induces apoptosis and the activation of the ASK1/JNK signaling pathway in human breast carcinoma MCF-7 cells [13]. On the contrary, our results indicated that the JNK signaling pathway was not activated in MG-63 cells after treatment with tomentosin. The Ras/Raf/ERK signaling pathway also plays a critical role in cell function and survival [21]. It was reported that sorafenib induced apoptosis and decreased pERK expression in HepG2 cells [22]. Interestingly, our results also indicated that the pERK expression level was significantly decreased in MG-63 cells after treatment with tomentosin. Taken together, our results suggested that tomentosin-induced apoptosis might be mediated by the ERK signaling pathway in MG-63 cells. 

ROS and the process of mitochondrial oxidative metabolism play an essential role in induction of apoptosis under both physiological and pathological conditions [23]. ROS can induce different pathologies, such as activation of apoptotic signaling pathways, disruption of intracellular redox homeostasis, and oxidative modifications of lipids, proteins, or DNA structure irreversibly [24]. In contrast to normal cells, cancer cells have high levels of intrinsic ROS because of an imbalanced mitochondrial oxidative metabolism process [23]. Intracellular ROS play crucial roles in the induction of apoptotic signaling pathways and regulation of cell proliferation in cancer cells [25].

Several studies have shown the anti-carcinogenic effect of natural compounds on human osteosarcoma cells. Kaempferol is a natural flavonoid, and it exhibits anti-cancer effects via endoplasmic reticulum stress and mitochondria-dependent apoptotic signaling pathways in human osteosarcoma U2OS cells [26]. Diosgenin is a plant steroid, which induces apoptosis via cell cycle arrest and activation of cyclooxygenase in the 1547 human osteosarcoma cell line [27]. Furthermore, ferulic acid, a ubiquitous phenolic acid found in corn, exerts potent anti-carcinogenic effects on human osteosarcoma MG-63 and 143B cells [28]. 

Peroxiredoxin-1 is a ROS scavenger protein and plays a crucial role in tumorigenesis [29,30]. Genistein is a flavonoid that induces apoptosis in human hepatocellular carcinoma SNU 449 cells via downregulation of peroxiredoxin-1 [31]. Furthermore, peroxiredoxin-1 has been shown to inhibit apoptosis via regulation of the apoptosis signal-regulating kinase-1 pathway in oral leukoplakia [32]. Our study suggested that downregulation of peroxiredoxin-1 induced cell apoptosis. Another study showed that peroxiredoxin-1 exerted anti-apoptotic effects in mouse models with oral precancerous lesions [33]. In this study, we found that the levels of the ROS scavenging protein peroxiredoxin-1 were decreased in MG-63 cells treated with tomentosin.

ROS is an important mediator of apoptosis in osteosarcoma cells. Deoxyelephantopin induces apoptosis via the induction of intracellular ROS in human osteosarcoma MG-63 and U2OS cells [34]. Moreover, nimbolide induces ROS production and apoptosis in MG-63 and U2OS cells [35]. Plumbagin induces apoptosis via intracellular ROS generation and mitochondrial apoptotic signaling pathways in MG-63 and U2OS cells [36]. However, previous studies could not elucidate the role of ROS in apoptosis. In this study, we examined the role of ROS in tomentosin-induced apoptosis.

To examine the role of intracellular ROS in tomentosin-induced apoptosis in MG-63 cells, we suppressed ROS production by treating cells with 10 mM NAC. Interestingly, our results showed that NAC treatment inhibited tomentosin-induced ROS generation in MG-63 cells (Figure 7c), which suggested that tomentosin induced intracellular ROS in MG-63 cells. Peroxiredoxin-1 is an antioxidant enzyme that reduces the level of hydrogen peroxide and alkyl hydroperoxides [37]. Interestingly, Western blot analysis showed that peroxiredoxin-1 levels were decreased after treatment with tomentosin. This result also supported the fact that tomentosin-induced apoptosis is closely associated with intracellular ROS. 

In this study, our results strongly supported the notion that tomentosin treatment increased intracellular ROS levels. Thus, we focused on the effects of tomentosin-induced ROS in osteosarcoma MG-63 cells. We observed that tomentosin treatment decreased the expression of peroxiredoxin-1, one of regulators of ROS production in MG-63 cells. Peroxiredoxin-1 is well known to play an important role in maintaining the ROS level for tumor development [38,39,40]. Furthermore, our results supported that tomentosin-induced intracellular ROS was an important mediator of the proliferation, migration, and apoptosis processes in MG-63 cells. We concluded that ROS might be a critical mediator of tomentosin-induced anti-cancer effect. Therefore, we mainly discuss the effect of ROS here. It is known that cancer cells have large amounts of ROS. As shown in Figure 4b, we compared the relative ROS levels after treatment with different concentrations of tomentosin and observed that the ROS level was gradually increased after treatment with increasing concentrations of tomentosin. Furthermore, as shown in Figure 4c,d, NAC treatment decreased the ROS level significantly in tomentosin-treated cells. It is well known that cell cycle arrest in the G2/M phase is usually mediated by p53. Even though we used MG-63 cells (p53 null) for this study, our results indicated that the percentage of cells in the G2/M phase in tomentosin-treated group was significantly higher than that in the NAC + tomentosin-treated group, which suggests that there might be another mediator of cell cycle arrest in MG-63 cells treated with tomentosin. It is known that FOXO3 can induce G2/M phase cell cycle arrest in cells through p27 and GADD45α expression [41,42,43,44]. Although FOXO3 was not reported to be involved in cell cycle arrest, Tang et al. showed that the key gene is p27, and treatment with lapatinib triggers G1 cell cycle arrest through both transcriptional and post-translational mechanisms in Her2-overexpressing breast cancer cell lines [45]. p27 can also mediate G2/M phase cell cycle arrest [46,47]. Thus, FOXO3 and p27 have the potential to participate in cell cycle arrest. Consistently, in our current study, we observed that MG-63 cells were under G2/M phase cell cycle arrest due to the increase in FOXO3 and p27 expression following treatment with tomentosin. Moreover, FOXO3 expression level in tomentosin-treated group was higher than that in NAC + tomentosin-treated group.

Many studies showed that FOXO3 plays an important role in cell cycle arrest. For example, Tiantian Sang et al. demonstrated that FOXO3 regulated cell cycle arrest through p27 upregulation [48]. Moreover, FOXO3 is known to be able to induce G1 and G2/M phase cell cycle arrest and apoptosis in breast cancer cells treated with paclitaxel [49]. Our results showed that tomentosin increased FOXO3 and p27 expression in MG-63 cells. Moreover, we demonstrated that tomentosin-induced ROS-up-regulated FOXO3 and p27 expression in MG-63 cells, which are consistent with the fact that FOXO3 is overexpressed in response to oxidative stress [50]. Taken together, we concluded that upregulation of FOXO3 may regulate G2/M phase cell cycle arrest through p27 upregulation after tomentosin treatment in MG-63 cells

To be categorized as an anti-cancer medicine, a compound should be tested as follows. First, the anti-carcinogenic effects, such as the cytotoxic effect of the compound should be tested in vitro. For this purpose, various kinds of cells are treated with the compound and its effects at the cellular level are investigated. Next, in vivo animal experiments are performed. For this, appropriate concentrations of the compound are injected into animals and its effects are investigated in various ways. Finally, clinical tests are performed to test the final medicine. Our study is closely related to the first step. Tomentosin is a natural compound derived from *I. viscosa* L. This plant has been recognized as a highly effective medicinal plant in Asia, including Korea. Although tomentosin has a high potential to be developed as an effective medicine, the effect of the compound in this plant was not well investigated at a cellular level. Therefore, we concluded that the effect of tomentosin on various cancer cells needs to be investigated. In this study, we focused on the effect of tomentosin on osteosarcoma MG-63 cells. Thus, our next approach will be in vivo animal tests using a xenograft and orthotopic mouse model to confirm the anti-cancer effects, such as anti-metastasis and apoptotic effects on osteosarcoma cells. We anticipate that our research will help in the development of new medicines to cure bone cancer.

## 4. Materials and Methods 

### 4.1. Reagents

Tomentosin was purchased from MCULE (Palo Alto, CA, USA) and dissolved in DMSO (Sigma-Aldrich, St. Louis, MO, USA). A 40 mM stock solution of tomentosin was stored at −20 °C. Antibodies for caspase-3, caspase-7, caspase-9, PARP, cleaved PARP, H2AX, γH2AX, Akt, JNK, ERK, pERK, Bcl-xl, Bcl-2, Bax, and peroxiredoxin-1 were purchased from Cell Signaling Technology (Danvers, MA, USA). Antibodies against caspase-8, pAkt (Ser473), pJNK, p38, pp38, were purchased from Santa Cruz Biotechnology (Santa Cruz, CA, USA). Anti-rabbit immunoglobulin G (IgG) and anti-mouse IgG were purchased from Cell Signaling Technology.

### 4.2. Cell Culture

Human osteosarcoma MG-63 cell line was purchased from the American Type culture Collection (ATCC, Manassas, VA, USA). The cells were incubated under standard conditions (37 °C, 5% CO_2_, and 95% humidity). MG-63 cells were cultured in Dulbecco’s modified Eagle’s medium (DMEM; Thermo Fisher Scientific, Grand Island, NY, USA) containing 10% heat-inactivated (56 °C and 30 min) fetal bovine serum (FBS; Sigma-Aldrich, St. Louis, Missouri, USA) and 1% penicillin/streptomycin antibiotics (Thermo Fisher Scientific, Grand Island, NY, USA).

### 4.3. Cell Proliferation Analysis Using MTT Assays

MG-63 cells were seeded in 96-well plates at a density of 5 × 10^3^ cells per well and incubated for 24 h. The cells were then treated with different concentrations of tomentosin (0, 10, 20, and 40 µM) and incubated for another 24 and 48 h. Following incubation, 20 µL of MTT dye (5 mg/mL) was added to each well and the cells were incubated for 2 h at 37 °C. The supernatants were then removed and cells were treated with formazan (dissolved in 200 µL of DMSO) and incubated in a shaker at room temperature (25 °C). After incubation, the absorbance was measured at 570 nm using a spectrophotometer (Molecular Devices, Mountain View, CA, USA).

### 4.4. Cell Proliferation Analysis Via Colony Formation and Cell Counting Assays

MG-63 cells were seeded in six-well plates at a density of 300 cells per well and incubated for 48 h. The cells were then treated with tomentosin (0 and 10 µM) for 48 h, following which the medium was replaced with fresh medium and MG-63 cells were incubated for 2 weeks under standard conditions. Next, the cells were washed with phosphate buffered saline (PBS) twice and fixed with 4% formaldehyde for 20 min at 4 °C. After fixation, the cells were washed with PBS twice and stained with 1% crystal violet (Sigma-Aldrich, St. Louis, Missouri, USA) dissolved in distilled water for 30 min and then the colonies were counted. To perform the cell counting assay, cells (3 × 10^3^) were seeded in six-well plates and incubated for 24 h under standard conditions. After incubation, the cells were treated with various concentrations of tomentosin (0, 20, and 40 µM) for 24 and 48 h. Cell numbers were then counted using a hemocytometer.

### 4.5. Cell Migration Analysis Using Wound Healing Assays

MG-63 cells were seeded in 24-well plates at a density of 5 × 10^4^ cells per well and incubated for 48 h. A wound was then created artificially by scraping with a yellow pipette tip and the cells were washed with PBS twice. The cells were treated with different concentrations of tomentosin (0, 20, and 40 µM). The wound was observed for 36 h at intervals of 12 h (0, 12, 24, and 36 h). The microscopy images of the cells were taken at 40× magnification by a microscope (CKX53; Olympus, Tokyo, Japan).

### 4.6. Boyden Chamber Assay

The invasive ability of MG-63 cells was evaluated using Boyden chamber invasion assays. FBS DMEM (1%) treated with different concentrations of tomentosin (0, 10, 20, and 40 µM) was loaded into the bottom part of the Boyden chamber. The cells were seeded to the upper part of the Boyden chamber at a density of 5 × 10^4^ cells in 50 µL of 0.1% FBS DMEM. A gelatin-coated membrane with 8 μm pore size was placed between the bottom and the upper parts of the Boyden chamber. After 3 h of incubation, the cells that invaded into the lower part of the chamber were stained using Diff-Quik staining solution (Sigma-Aldrich, St. Louis, Missouri, USA). The number of invaded MG-63 cells was counted.

### 4.7. Zymography Analysis

The activity of MMPs, especially gelatinases, was measured using the gelatin zymography assay. MG-63 cells were seeded in 100 mm cell culture dishes (1 × 10^6^ cells) and incubated under standard conditions (37 °C, 5% CO_2_, and 95% humidity) for 24 h. The cells were then treated with different concentrations of tomentosin (0, 10, and 20 µM) in medium, which contained 0.1% FBS, and incubated for 48 h under standard conditions. After incubation, the cultured medium was concentrated using an Amicon Ultra-15 filter (Millipore, Billerica, MA, USA). Protein concentration was measured using Bradford protein assays. Concentrated media samples were mixed with zymography sample buffer and separated by gelatin-containing 8% acrylamide gels. The gel was incubated in developing buffer overnight. After that, the gel was stained with a Coomassie staining buffer (Sigma-Aldrich, St. Louis, Missouri, USA). 

### 4.8. Cell Cycle Analysis Using Flow Cytometry

MG-63 cells were seeded at a density of 2 × 10^5^ cells in 60 mm cell culture dishes and incubated for 48 h. The cells were then treated with different concentrations of tomentosin (0, 10, 20, and 40 µM) for 48 h, following which they were harvested by trypsinization and fixed in ice-cold 70% ethanol overnight at −20 °C. After fixation, the cells were centrifuged at 1350 rpm for 5 min and incubated with PI working solution (Sigma-Aldrich; 50 µg/mL PI and 200 µg/mL RNaseA) for 30 min at 37 °C. Cell cycle distribution analysis was performed using flow cytometry (Beckman Coulter, Brea, CA, USA).

### 4.9. Apoptosis Analysis Using Flow Cytometry

The percentage of apoptotic cells was measured using the FITC Annexin V apoptosis detection kit (BD Biosciences, Franklin Lakes, NJ, USA). MG-63 cells were seeded in six-well plates at a density of 3 × 10^3^ cells per well and incubated for 24 h. The cells were then treated with different concentrations of tomentosin (0, 10, 20, and 40 µM) for 24 and 48 h. Next, cells were washed with PBS and suspended in 1× binding buffer. The cells were stained with FITC-labeled annexin V and PI for 15 min at room temperature in the dark. After staining, cells were analyzed using flow cytometry (Beckman Coulter, Brea, CA, USA).

### 4.10. DNA Fragmentation Analysis Using TUNEL Assays 

We observed the fluorescence of apoptotic cells by TUNEL assay using the fluorometric TUNEL detection system (Promega, Madison, WI, USA). MG-63 cells were seeded in six-well plates at a density of 2.5 × 10^5^ cells per well and incubated for 24 h. The cells were then treated with different concentrations of tomentosin (0, 10, 20, and 40 µM) for 24 and 48 h. Next, the cells were fixed with 4% formaldehyde for 20 min at 4 °C and permeabilized using 0.5% triton X-100 for 10 min at room temperature. Following that, the cells were treated with 50 µL TdT enzyme buffer and incubated for 1 h at 37 °C. The cell nucleus was stained using a Hoechst stain solution (Sigma-Aldrich). One microliter of Hoechst stain solution was dissolved in 2 mL of PBS. Labeled strand breaks were observed by fluorescence microscopy (CKX53; Olympus, Shinjuku, Tokyo, Japan).

### 4.11. Western Blot Analysis

MG-63 cells were treated with different concentrations of tomentosin (0, 10, 20, and 40 µM) for 48 h and lysed using a RIPA buffer (Sigma-Aldrich). The cell lysates were centrifuged at 17,000 rpm for 15 min and the supernatant was collected. The concentrations of cell lysate protein were measured using a Qubit™ Fluorocytometer (Invitrogen, Carlsbad, CA, USA). Total protein (15 µL) was separated on 10% sodium dodecyl sulfate-polyacrylamide gels at 100 V for 24 h and transferred onto a 0.45 µm nitrocellulose blotting membrane (GE Healthcare, Little Chalfont, United Kingdom) at 50 V for 2 h. The membranes were blocked with 3% bovine serum albumin (Bovogen, 12 Williams Ave, Keilor East VIC 3033, Australia) for 1 h at room temperature. The membranes were incubated with primary antibodies overnight at 4 °C, washed with TBS-T, and then incubated with a secondary antibody (Cell Signaling Technology). The membranes were washed with TBS-T again and chemiluminescence was detected using ECL (iNtRON Biotechnology, South Korea) and Chemidoc detection system (Bio-Rad, Hercules, CA, USA).

### 4.12. ROS Detection by Flow Cytometry

Intracellular ROS level was measured using the stable nonpolar dye DCF-DA, which readily diffuses into the cells. MG-63 cells were treated with different concentrations of tomentosin (0, 10, 20, and 40 µM) for 48 h and then incubated at 37 °C with 20 μM of DCF-DA for 30 min. After incubation, the ROS level was measured by flow cytometry (Beckman Coulter). For investigating the ROS dependency of tomentosin for its anti-cancer activity, MG-63 cells were treated with either 40 μM of tomentosin, 10 mM of NAC, or 40 μM of tomentosin + 10 mM of NAC for 48 h.

### 4.13. siRNA Transfection

Scramble control siRNA and siRNA against FOXO3 were purchased from Santa Cruz Biotechnology (Santa Cruz, CA, USA). For transfection with siRNA, cells were transfected with scramble control siRNA or siRNA against FOXO3 using Lipofectamine 2000 transfection reagent (Thermo Scientific, Rockford, IL) according to the manufacturer’s protocol.

### 4.14. Statistical Analysis

Data are representative of three independent experiments. Student’s *t*-test and one-way analysis of variance (ANOVA), followed by a Bonferroni post-hoc test, were performed to statistically analyze the data, and a *p*-value of < 0.05 was considered statistically significant. 

## 5. Conclusions

In conclusion, our results demonstrated that tomentosin induced apoptosis in human osteosarcoma MG-63 cells and increased the levels of intracellular ROS, which are important mediators of apoptosis in MG-63 cells. 

## Figures and Tables

**Figure 1 ijms-20-01508-f001:**
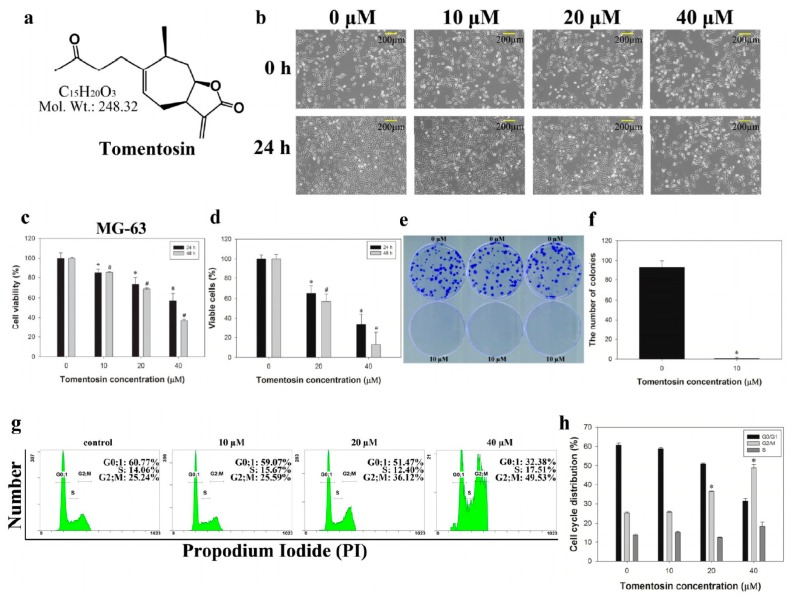
Cell cytotoxicity assay of MG-63 cells treated with tomentosin. (**a**) Chemical structure of tomentosin. (**b**) Morphological changes in tomentosin-treated MG-63 cells. (**c**) Cell viability was determined using an MTT assay. **p* and #*p* < 0.05 compared with control cells. (**d**) The relative cell survival rate was determined by cell counting assay. **p* and #*p* < 0.05 compared with control cells. (**e**) Colony formation assay of tomentosin-treated MG-63 cells. The cells were treated with 10 µM of tomentosin for 24 h; subsequently, colonies were allowed to grow for 10 days and the number of colonies was counted. (**f**) Colony formation assay results were analyzed using the Student’s *t*-test. **p* < 0.05 compared with control cells. (**g**) Cell cycle analysis of MG-63 cells. The cells were treated with 0, 10, 20, and 40 µM of tomentosin for 48 h and stained with propodium iodide (PI). After staining, the cells were analyzed using flow cytometry. The distribution and percentage of cells in the G0, S, and G2/M phase of the cell cycle were evaluated. (**h**) The results were analyzed statistically using the Student’s *t*-test. **p* < 0.05 compared with control cells.

**Figure 2 ijms-20-01508-f002:**
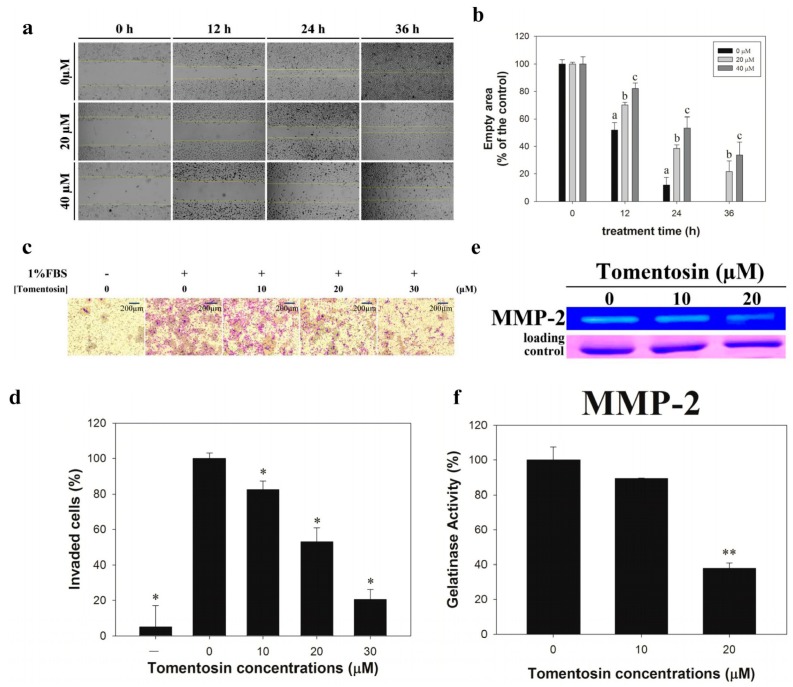
Inhibitory effects of tomentosin on migration and invasion of MG-63 cells. (**a**) A wound healing assay was performed in MG-63 cells treated with tomentosin. The cells were treated with different concentrations of tomentosin (0, 20, and 40 µM) and the wound area was observed for 36 h at intervals of 12 h. (**b**) The results were analyzed using the Student’s *t*-test. a,b,c: *p* < 0.05 compared with control cells. (**c**) Invasion of MG-63 cells treated with different concentrations of tomentosin (0, 10, 20, and 40 µM) was analyzed using the Boyden chamber assay. (**d**) The results were analyzed statistically using the Student’s *t*-test. **p* < 0.05 compared with the control (1% FBS treated only) cells. (**e**) Proteolytic activity of MMP-2 decreased in MG-63 cells after treatment with tomentosin (0, 10, and 20 μM) for 48 h. (**f**) The results were analyzed statistically using the Student’s *t*-test. ***p* < 0.01 compared with control cells.

**Figure 3 ijms-20-01508-f003:**
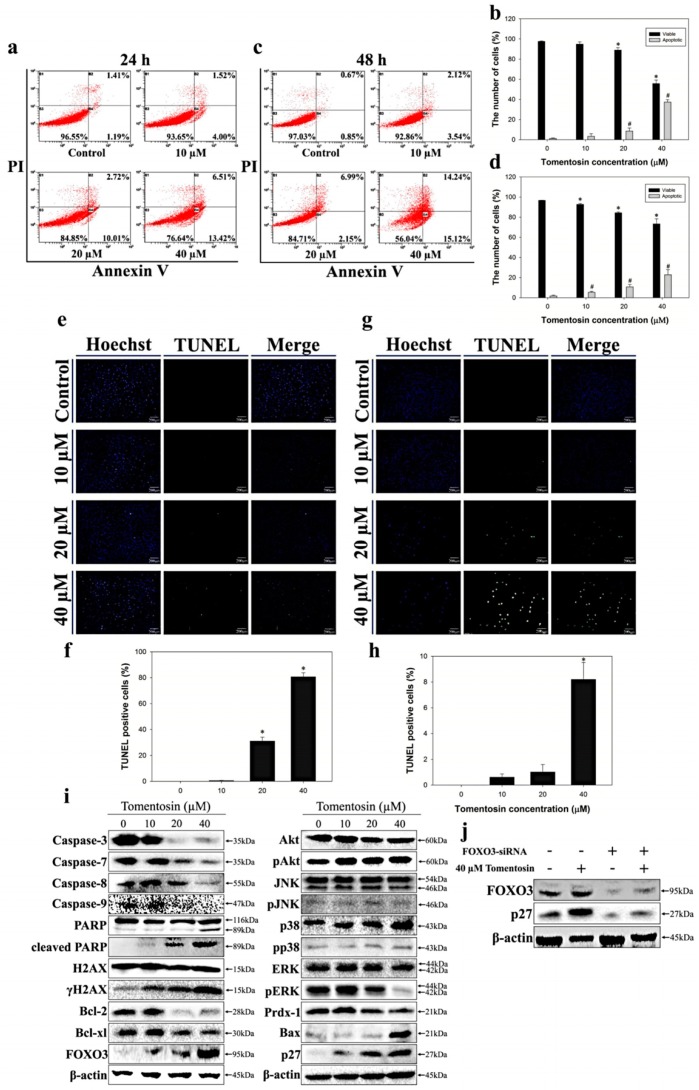
Apoptosis induced by tomentosin in MG-63 cells. Fluorescence-activated cell sorting (FACS) analysis was undertaken of MG-63 cells treated with tomentosin to evaluate apoptosis. The cells were treated with different concentrations of tomentosin (0, 10, 20, and 40 µM) for 24 and 48 h and then analyzed using flow cytometry. Scatter plots represent the distribution of annexin V/PI staining for control and tomentosin-treated MG-63 cells. The cells were categorized as “viable” (lower left), “early apoptotic” (lower right), and “late apoptotic” (upper right). Flow cytometry analysis of cells after treatment with tomentosin for 24 (**a**) and 48 h (**c**). Quantitative analysis of the viable, early apoptotic, and late apoptotic cells after treatment with tomentosin for 24 (**b**) and 48 h (**d**) using FACS analysis. **p* and #*p* < 0.05 compared with control cells. Detection of apoptosis using a TUNEL assay in MG-63 cells treated with tomentosin for 24 (**e**) and 48 h (**g**). The cells were treated with different concentrations of tomentosin (0, 10, 20, and 40 µM) for 24 and 48 h and then analyzed by the TUNEL detection system. Representative images showing TUNEL-positive nucleus (green color) (100× magnification). Quantitative analysis of the TUNEL-positive cells after treatment with tomentosin for 24 (**f**) and 48 h (**h**). The results were analyzed using the Student’s *t*-test. **p* < 0.05 compared with control cells. (**i**) Western blot analysis after treatment of MG-63 cells with tomentosin. The cells were treated with different concentrations of tomentosin (0, 10, 20, and 40 µM) for 48 h and Western blot analysis was performed with specific antibodies as indicated. Expression levels of caspase-3, caspase-7, caspase-8, caspase-9, PARP, cleaved PARP, H2AX, γH2AX, Bcl-2, Bcl-xl, FOXO3, Akt, pAKT, JNK, pJNK, p38, pp38, ERK, pERK, peroxiredoxin-1, Bax, and p27 were evaluated. β-actin was used as a loading control. (**j**) Western blot analysis of FOXO3 and p27 expression level in MG-63 cells transfected with control siRNA or FOXO3 siRNA followed by tomentosin treatment. β-actin was used as a loading control. Scale bar = 200 μm in (**e**,**g**).

**Figure 4 ijms-20-01508-f004:**
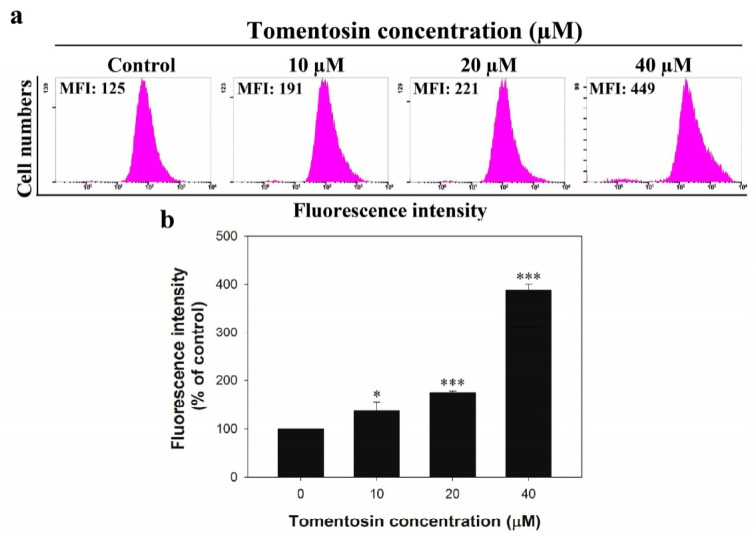
Measurement of ROS level in MG-63 cells after treatment with tomentosin. (**a**) The cells were treated with different concentrations of tomentosin (0, 10, 20, and 40 µM) for 48 h and DCF-DA staining was performed to measure fluorescence intensity. MFI refers to mean fluorescence intensity. (**b**) Statistical analysis of fluorescence intensity. **p* < 0.05 and ****p* < 0.001 compared with control cells. (**c**) The cells were treated with 10 mM of NAC and/or 40 µM of tomentosin for 48 h and DCF-DA staining was performed to measure fluorescence intensity. (**d**) Statistical analysis of fluorescence intensity. **p* < 0.05 compared with control cells and #*p* < 0.05 compared with tomentosin-treated cells.

**Figure 5 ijms-20-01508-f005:**
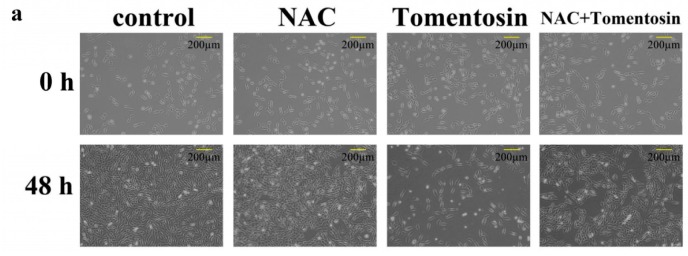
Cytotoxic effects of tomentosin-induced intracellular ROS in MG-63 cells. The cells were treated with 10 mM of NAC and/or 40 µM of tomentosin for 48 h. (**a**) Morphological changes in MG-63 cells treated with 10 mM of NAC and/or 40 µM of tomentosin. (**b**) Cell viability was determined using an MTT assay. (**c**) The relative cell survival rate was determined using a cell counting assay. (**d**) Cell cycle analysis of MG-63 cells. (**e**) The results were analyzed by the Student’s *t*-test. **p* < 0.05 compared with control cells and #*p* < 0.05 compared with tomentosin-treated cells.

**Figure 6 ijms-20-01508-f006:**
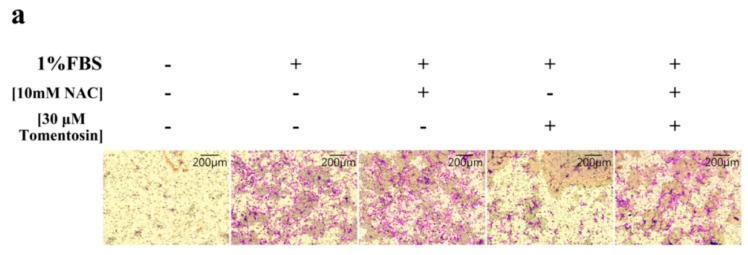
Inhibitory effects of tomentosin on the migration and invasion of MG-63 cells after treatment with 10 mM of NAC and/or 40 µM of tomentosin for 48 h. (**a**) Invasion of MG-63 cells was analyzed using the Boyden chamber assay. (**b**) The results were analyzed using the Student’s *t*-test. **p* < 0.05 compared with control cells and #*p* < 0.05 compared with tomentosin-treated cells. (**c**) Proteolytic activity of MMP-2 was analyzed by zymography assay. (**d**) The results were analyzed using the Student’s *t*-test. **p* < 0.05 compared with control cells and #*p* < 0.05 compared with tomentosin-treated cells.

**Figure 7 ijms-20-01508-f007:**
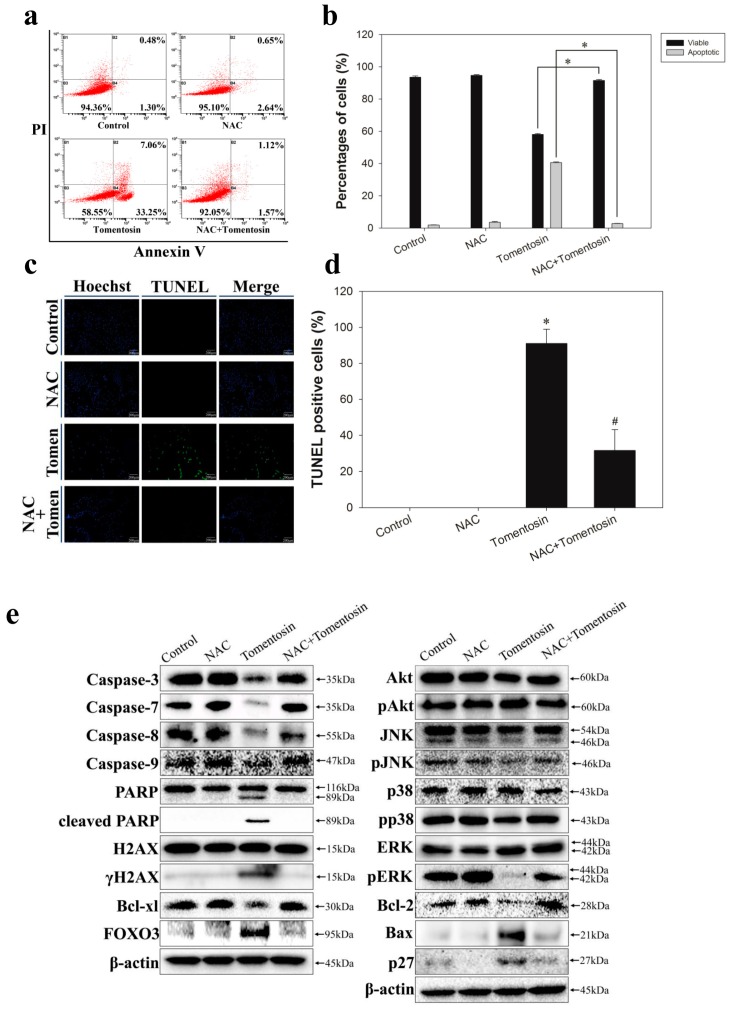
Effect of tomentosin-induced intracellular ROS on MG-63 cell apoptosis. (**a**) Flow cytometry analysis. (**b**) Quantitative analysis of viable, early apoptotic, and late apoptotic cells. **p* < 0.05 compared with tomentosin-treated cells. (**c**) Detection of apoptosis using a TUNEL assay in MG-63 cells. Representative images showing TUNEL-positive nucleus (green color) (100× magnification). (**d**) Quantitative analysis of the TUNEL-positive cells. **p* <0.05 compared with control cells and #*p* < 0.05 compared with tomentosin-treated cells. (**e**) Western blot analysis was performed with specific antibodies as indicated. Expression levels of caspase-3, caspase-7, caspase-8, caspase-9, PARP, cleaved PARP, H2AX, γH2AX, Bcl-xl, FOXO3, Akt, pAKT, JNK, pJNK, p38, pp38, ERK, pERK, Bcl-2, Bax, and p27 were evaluated. β-actin was used as a loading control.

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
