# Peer review of "Tomentosin Displays Anti-Carcinogenic Effect in Human Osteosarcoma MG-63 Cells via the Induction of Intracellular Reactive Oxygen Species"

_ijms, 2019, doi:10.3390/ijms20061508_

Reviewer 1 Report

The paper by Lee and colleagues presents a study on the effect of Tometosin in a human osteosarcoma cell line. The mechanism of action of this compound, based on the induction of intracellular ROS, is also presented.
The manuscript is well written, clear and concise, and scientifically sound. The results are robust and clearly presented. the experimental plan is appropriate.
Minor points:
-Figure 3, panels e and g,  and 7 panel c are difficult to read due to their very small dimensions. Authors should provide larger images as supplementary materials.
-Authors should provide a comment on the OS cell line choosen. Why MG-63 and not other OS cell lines such as SaOS-2 or U-2OS ?

Author Response

Reviewer 1

The paper by Lee and colleagues presents a study on the effect of Tomentosin in a human osteosarcoma cell line. The mechanism of action of this compound, based on the induction of intracellular ROS, is also presented.

The manuscript is well written, clear and concise, and scientifically sound. The results are robust and clearly presented. The experimental plan is appropriate.

Minor points:

- Figure 3, panel e and g, and 7 panel c are difficult to read due to their very small dimensions. Authors should provide larger images as supplementary materials.

Answer: We really appreciate these constructive comments. We revised Figure 3 and 7 images according to reviewer’s requests.

-Authors should provide a comment on the OS cell line chosen. Why MG-63 and not other OS cell lines such as SaOS-2 or U-2OS?

Answer: We really appreciate these constructive comments. We were trying to select MG-63 osteosarcoma cell lines for this specific study because we wanted to investigate the possibility of tomentosin as the novel therapeutic options in the p53 null osteosarcoma patient. MG-63 cells are known to have no functional p53 (p53 null) and thus is good model cell line for the development of the novel therapeutic treatment for osteosarcoma patients with p53 null status. We incorporated the above paragraph into the discussion part.

Reviewer 2 Report

Testing novel therapeutic agents to improve the treatment of osteosarcomas is an important purpose. The rational to test Tomentosin is that it was already widely used because it’s anti-inflammatory, anti-bacterial and anti-cancer (anti-proliferative) action.   

The manuscript is carefully written and the techniques are well described although some precisions should be given.

The main weaknesses of the study is that the authors conclude that Tomentosin acts both on cell survival, proliferation, migration, invasiveness and also induces massive cell death.  Since 80 % of the cells are TUNEL positive and therefore are in advanced apoptosis (whereas only a small proportion is annexinV positive) how could be expected that these cells still proliferate, migrate and invade.

In another hand, how could the apoptotic cells be in G2/M whereas no cells were in sub-G0. The gating strategy for FACS analyses should be shown. Is it sure that the cells analyzed in G2/M were not the 20% cells that are not apoptotic after the treatment? May be cells that did not respond to Tomentosin and were selected by the treatment? It is an important point because it was discussed that ROS induction was associated with G2M arrest. The effect of ROS on cell survival, DNA damages and proliferation arrest are well known. Prolonged arrest in G2 may end up with apoptosis but it is dependent on p53 and MG63 cells have no functional p53. So the experimental conditions should be re-examined and more detailed conclusions and discussion should be provided.

The results clearly show that the pro-apoptotic function of Tomentosin mainly depends on ROS induction although it may be in contradiction with the fact that cancer cells are already rich in ROS. The authors should therefore also discuss about the specificity of the actions of Tomentosin in osteosarcoma cells.

What about the possible effects on healthy surrounding cells?

Although the rational for the study was the need of novel osteosarcoma treatments, the authors did not discuss how this compound could be a medicine in patients with bone tumors or be part of a chemotherapeutic protocol.

Minor corrections

The sentence in lines 42-44 is very confusing. It suggests that the biology of osteosarcomas is very well understood and that previous knowledges support the development of novel treatment with natural chemical compounds. This should be more precisely explained.  

In fig. 1b, the size of the photos and the magnification are really too small to observe any morphological changes of the cells.

It is actually expected (and therefore not so interesting) that the ROS inhibitor, NAC, inhibits ROS (lines 320-321).

The authors tended to repeat the results in the discussion instead of discussing them.

How was chosen the cell line (MG63) and the doses of Tomentosin?

Student’s t-tests are not appropriated when performing multiple comparisons.

Author Response

Reviewer 2

Testing novel therapeutic agents to improve the treatment of osteosarcomas is an important purpose. The rational to test Tomentosin is that it was already widely used because it’s anti-inflammatory, anti-bacterial and anti-cancer (anti-proliferative) action.

The manuscript is carefully written and the techniques are well described although some precisions should be given.

-The main weaknesses of the study are that the authors conclude that Tomentosin acts both on cell survival, proliferation, migration, invasiveness and also induces massive cell death. Since 80 % of the cells are TUNEL positive and therefore are in advanced apoptosis (whereas only a small proportion is annexinV positive) how could be expected that these cells still proliferate, migrate and invade.

Answer: We really appreciate these constructive comments. When treated the cells with the relatively low concentrations of tomentosin (0, 10, and 20 µM), apoptotic cell rates were low. However, low dose of tomentosin decreased migration ability (migration assay) and invasion ability significantly (invasion assay). Moreover, gelatinase activity is significantly decreased in 20 µM of tomentosin-treated group. These results suggested that tomentosin suppressed migration and invasion ability in the condition of the low concentration tomentosin. When the cells were treated with 40 µM of tomentosin, however, the proportion of TUNEL-positive cells was much higher than other groups. These results suggested that tomentosin induced apoptosis and cell death when the cells were treated with the high concentration of tomentosin. Taken together, our results suggested that tomentosin suppressed migration/invasion ability and subsequently induced apoptosis in MG-63 cells depending on the treatment dose. We incorporated the above paragraph into the discussion part.

-In another hand, how could the apoptotic cells be in G2/M whereas no cells were in sub-G0. The gating strategy for FACS analyses should be shown. Is it sure that the cells analyzed in G2/M were not the 20% cells that are not apoptotic after the treatment? May be cells that did not respond to Tomentosin and were selected by the treatment? It is an important point because it was discussed that ROS induction was associated with G2M arrest. The effect of ROS on cell survival, DNA damages and proliferation arrest are well known. Prolonged arrest in G2 may end up with apoptosis but it is dependent on p53 and MG63 cells have no functional p53. So the experimental conditions should be re-examined and more detailed conclusions and discussion should be provided.

Answer: We really appreciate these constructive comments. As reviewer mentioned, it is well known that G2/M phase cell cycle arrest was usually mediated by p53. Even though we used MG-63 cells (p53 null) for this study, our results indicated that the percentage of G2/M phase in tomentosin-treated group was significantly higher than tomentosin + NAC-treated group, which suggest that there might be another mediator for cell cycle arrest in MG-63 cells treated with tomentosin. It is known that FOXO3 can induce the G2/M phase cell cycle arrest in cells through p27 and GADD45α expression (J Biol Chem. 277(30):26729-32. FOXO forkhead transcription factors induce G(2)-M checkpoint in response to oxidative stress, J Immunol. 189(10):4748-58. APRIL binding to BCMA activates a JNK2-FOXO3-GADD45 pathway and induces a G2/M cell growth arrest in liver cells, Oncol Rep. 9(1):103-8. Casticin induces growth suppression and cell cycle arrest through activation of FOXO3a in hepatocellular carcinoma, Oncotarget. 6(42):44819-31. Tumor suppressive effect of PARP1 and FOXO3A in gastric cancers and its clinical implications.). Although FOXO3 was not mentioned to be involved in cell cycle arrest, Tang et al investigated that the key gene is p27 where Lapatinib trigger G1 cell cycle arrest through both transcriptional and post-translational mechanisms in Her2-over-expressing breast cancer cell line (Cell Cycle. 12(16):2665-74. Lapatinib, one of tyrosine kinase inhibitors, induces p27(Kip1)-dependent G₁ arrest through both transcriptional and post-translational mechanisms. Tang L1, Wang Y, Strom A, Gustafsson JÅ, Guan X.). Thus, FOXO3 has a potential to participate in cell cycle arrest and consistently in our current study, we have observed that MG-63 cells were under G2/M phase cell cycle arrest by the increase in FOXO3 and p27 expression with treatment of tomentosin. Moreover, FOXO3 expression level in tomentosin-treated group was higher than tomentosin + NAC-treated group. We incorporated the above paragraph into the discussion part.

The results clearly show that the pro-apoptotic function of Tomentosin mainly depends on ROS induction although it may be in contradiction with the fact that cancer cells are already rich in ROS. The authors should therefore also discuss about the specificity of the actions of Tomentosin in osteosarcoma cells.

Answer: We really appreciate these constructive comments. In this study, our results strongly supported that tomentosin increased intracellular ROS. Thus, we focused on the effects of ROS induced by tomentosin in osteosarcoma MG-63 cells. We observed that tomentosin treatment decreased the expression of peroxiredoxin-1, one of regulators of ROS level in MG-63 cells. Peroxiredoxin-1 is well-known to play an important role in maintaining ROS level for tumor development (Cell Chem Biol. 9456(18)30435-5. Frenolicin B Targets Peroxiredoxin 1 and Glutaredoxin 3 to Trigger ROS/4E-BP1-Mediated Antitumor Effects. Targeting peroxiredoxin 1 impairs growth of breast cancer cells and potently sensitises these cells to prooxidant agents. Br. J. Cancer. 119: 873-884. Thioredoxins, glutaredoxins, and peroxiredoxins–molecular mechanisms and health significance: from cofactors to antioxidants to redox signaling. Antioxid. Redox Signal. 19: 1539-1605). Furthermore, our results supported that intracellular ROS induced by tomentosin was important mediator of proliferation, migration, and apoptosis process in MG-63 cells. We concluded that ROS may be a specific mediator in tomentosin-induced anti-cancer effect. Therefore, we discussed mainly about the effect of ROS in discussion section. We also agreed with your opinion that cancer cells have the considerable amount of ROS. As shown in Figure 4B, we compared the relative ROS levels after treatment of different concentrations of tomentosin and observed that ROS level was gradually increased after treatment of the increasing concentration of tomentosin. Furthermore, as shown in Figure 4C and D, NAC treatment decreased ROS level significantly in tomentosin-treated cells. We incorporated the above paragraph into the discussion part.

What about the possible effects on healthy surrounding cells?

Answer: We really appreciate these constructive comments. Although in the current manuscript, we did not present the results of the possible effect in the normal cells by the tomentosin treatment, in fact, we did test the cytotoxic effect of tomentosin against two kind of normal cell lines (WRL63 and CCL13). Unfortunately, we observed that the high dose (40 µM) of tomentosin have the similar cytotoxicity to WRL63 and CCL13 cell lines. So, we are focusing on the anti-migration and anti-invasion effect of tomentosin in MG-63 cells since the effective dose of tomentosin was relatively low (10 µM) and 10 µM of tomentosin did not induce the cell death in WRL63 and CCL13 cell lines. However, we just tested the anti-cancer effect of tomentosin at the level of in vitro cellular assay and we should confirm the anti-cancer effect of tomentosin using in vivo xenograft and orthotopic mouse model.

Although the rational for the study was the need of novel osteosarcoma treatments, the authors did not discuss how this compound could be a medicine in patients with bone tumors or be part of a chemotherapeutic protocol.

Answer: We really appreciate these constructive comments. To be qualified as an anti-cancer medicine, one compound should be tested as follows. First, the anti-carcinogenic effect such as the cytotoxic effect of the compound should be tested in vitro. Various kinds of cells are treated with the compound and investigate the effects in cellular level. Next, in vivo animal experiments should be performed. Appropriate concentrations of compound were injected in animals and investigate the effects in various ways. Finally, clinical tests were performed to be a final medicine. Our study is closely related to first step. Tomentosin is a natural compound derived from I. viscosa L. This plant has been highly recognized as an effective medicinal plant in Asia including Korea. Although tomentosin has a high potential to be developed as an effective medicine, the effect of compound involved in this plant was not investigated well in cellular level. Therefore, we concluded that it needs to be investigated about the effect of tomentosin in various cancer cells. In this study, we focused on the effect of tomentosin in osteosarcoma MG-63 cells. Thus, our next approach will be in vivo animal test using xenograft and orthotopic mouse model to confirm the anti-cancer effect such as anti-metastasis and apoptosis of osteosarcoma. We anticipate that our research helps to develop new medicine to cure bone cancer. We incorporated the above paragraph into the discussion part.

Minor corrections

The sentence in lines 42-44 is very confusing. It suggests that the biology of osteosarcomas is very well understood and that previous knowledges support the development of novel treatment with natural chemical compounds. This should be more precisely explained.  

Answer: We really appreciate these constructive comments. We deleted line 42-44 according to reviewer’s requests.

In fig. 1b, the size of the photos and the magnification are really too small to observe any morphological changes of the cells.

Answer: We really appreciate these constructive comments. We also agreed that it is difficult to observe the morphological changes detail in images. However, we tried to show that the number of cells decreased after treatment of tomentosin and in fact, our results (Figure 1B) indicated that the number of cells decreased with dose dependent manner.

It is actually expected (and therefore not so interesting) that the ROS inhibitor, NAC, inhibits ROS (lines 320-321).

Answer: We really appreciate these constructive comments. We deleted line 320-321 according to reviewer’s requests.

The authors tended to repeat the results in the discussion instead of discussing them.

Answer: We really appreciate these constructive comments. We also agreed that some results are repeated in discussion part. To draw conclusion in this study, we feel that we need to interpret and organize results in discussion section. Also, we incorporated the revised paragraph into the discussion part according to reviewers’ comments.

How was chosen the cell line (MG63) and the doses of Tomentosin?

Answer: We really appreciate these constructive comments. We were trying to select MG-63 osteosarcoma cell lines for this specific study because we wanted to investigate the possibility of tomentosin as the novel therapeutic options in the p53 null osteosarcoma patient. MG-63 cells are known to have no functional p53 (p53 null) and thus is good model cell line for the development of the novel therapeutic treatment for osteosarcoma patients with p53 null status. In this study, we investigated the anti-carcinogenic effect of tomentosin in human osteosarcoma MG-63 cells. When we treated the cells with the higher dose than 40 µM of tomentosin, the cytotoxic effect is too strong and almost cells died. Therefore, we concluded that 40 µM is a maximum concentration in this study. We incorporated the above paragraph into the discussion part.

Student’s t-tests are not appropriated when performing multiple comparisons.

Answer: We really appreciate these constructive comments. We performed One-way ANOVA test when performing multiple comparisons.

Round  2

Reviewer 2 Report

It is not possible to extrapolate that the increased FOXO3a is responsible for G2M arrest from the results of this study based only the results showing that FOXO3a and p27 were both increased and that p27 controlled G1 arrest in a very different context. FOXO3a was not directly shown to be involved in G2M arrest.

The text has to be carefully checked because of new sentences especially in the discussion that are very confusing or need English correction. There are also sentences that are repeated twice ( lines 274-289 and 284-289).

Author Response

March 19th, 2019

Dear International Journal of Molecular Sciences

Thank you for your letter from March 13th, 2019, inviting us to submit a revised version of manuscript: Manuscript ID: ijms-453453 “Tomentosin Displays Anti-carcinogenic Effect in Human Osteosarcoma MG-63 Cells via Induction of Intracellular ROS” by Lee et al. We appreciate the careful review of the manuscript by the reviewers. In response, we have incorporated all suggested changes and revisions. We feel that the suggested revisions have significantly strengthened the paper. Please find a point by point response to each of the concerns posed by the reviewer below. Revisions introduced into the text are in red (we use track change methods).

Thank you for your kind consideration.

Sincerely yours,

Prof. See-Hyoung Park, PhD

Department of Biological and Chemical Engineering

Hongik University

Sejong, Korea.

Reviewer 2

It is not possible to extrapolate that the increased FOXO3a is responsible for G2M arrest from the results of this study based only the results showing that FOXO3a and p27 were both increased and that p27 controlled G1 arrest in a very different context. FOXO3a was not directly shown to be involved in G2M arrest.

Answer: We really appreciate these constructive comments. We agreed with your opinion. To verify the role of FOXO3 for G2/M cell cycle arrest in MG-63 cells treated with tomentosin, we performed Western blotting analysis of FOXO3 and p27 expression level in MG-63 cells transfected with control siRNA or FOXO3 siRNA followed by tomentosin treatment. As shown in Figure 3j, after tomentosin treatment for 24 h, compared to DMSO control both of FOXO3 and p27 expression level was increased in MG-63 cells transfected with control siRNA but neither of FOXO3 nor p27 expression level was not changed in MG-63 cells transfected with siRNA against FOXO3. This result suggests that tomentosin may induce p27-mediated G2/M cell cycle arrest with a FOXO3-dependent manner. / Many studies showed that FOXO3 plays an important role in cell cycle arrest. For example, Tiantian Sang et al demonstrated that FOXO3 regulated cell cycle arrest through p27 upregulation (Overexpression or Silencing of FOXO3a Affects Proliferation of Endothelial Progenitor Cells and Expression of Cell Cycle Regulatory Proteins, Tiantian Sang et al, 2014, PLOS ONE). Moreover, FOXO3 is known to be able to induce G1 and G2/M phase cell cycle arrest and apoptosis in breast cancer cell treated with paclitaxel. (FoxO3a Transcriptional Regulation of Bim controls Apoptosis in Paclitaxel-treated Breast Cancer Cell lines, Andrew Sunters et al, 2003, Journal of Biological Chemistry). Our results showed that tomentosin increased FOXO3 and p27 expression in MG-63 cells and  we demonstrated that tomentosin-induced ROS up-regulated FOXO3 and p27 expression in MG-63 cells, which are consistent with the fact that FOXO3 is overexpressed in response to oxidative stress (Hydrogen peroxide induced impairment of endothelial progenitor cell viability is mediated through a FoxO3a dependent mechanism, Fel Wang et al, 2013, Microvascular Research). Taken together, we concluded that upregulation of FOXO3 may regulate G2/M phase cell cycle arrest through p27 upregulation after tomentosin treatment in MG-63 cells. We incorporated the above paragraph into the discussion part.

The text has to be carefully checked because of new sentences especially in the discussion that are very confusing or need English correction. There are also sentences that are repeated twice (lines 274-289 and 284-289).

Answer: We really appreciate these constructive comments. We agreed that our new sentences in the discussion part are not natural. Therefore, we removed repeated sentences and ask professional institution to revise our manuscript. We incorporated the English editing certificate at the last page of manuscript.

Round  3

Reviewer 2 Report

I still believe that you should better explore which cell population is arrested in G2/M and whether this population was actually those which enter apoptosis.